# Targeting GPCRs and Their Signaling as a Therapeutic Option in Melanoma

**DOI:** 10.3390/cancers14030706

**Published:** 2022-01-29

**Authors:** Jérémy H. Raymond, Zackie Aktary, Lionel Larue, Véronique Delmas

**Affiliations:** 1Institut Curie, Université PSL, CNRS UMR3347, Inserm U1021, Normal and Pathological Development of Melanocytes, 91400 Orsay, France; jeremy.raymond@curie.fr (J.H.R.); zackie.aktary@curie.fr (Z.A.); veronique.delmas@curie.fr (V.D.); 2Université Paris-Saclay, CNRS UMR3347, Inserm U1021, Signalisation Radiobiologie et Cancer, 91400 Orsay, France

**Keywords:** G-protein-coupled receptor, mouse models, skin cancer, UVR, drug, network

## Abstract

**Simple Summary:**

Sixteen G-protein-coupled receptors (GPCRs) have been involved in melanogenesis or melanomagenesis. Here, we review these GPCRs, their associated signaling, and therapies.

**Abstract:**

G-protein-coupled receptors (GPCRs) serve prominent roles in melanocyte lineage physiology, with an impact at all stages of development, as well as on mature melanocyte functions. GPCR ligands are present in the skin and regulate melanocyte homeostasis, including pigmentation. The role of GPCRs in the regulation of pigmentation and, consequently, protection against external aggression, such as ultraviolet radiation, has long been established. However, evidence of new functions of GPCRs directly in melanomagenesis has been highlighted in recent years. GPCRs are coupled, through their intracellular domains, to heterotrimeric G-proteins, which induce cellular signaling through various pathways. Such signaling modulates numerous essential cellular processes that occur during melanomagenesis, including proliferation and migration. GPCR-associated signaling in melanoma can be activated by the binding of paracrine factors to their receptors or directly by activating mutations. In this review, we present melanoma-associated alterations of GPCRs and their downstream signaling and discuss the various preclinical models used to evaluate new therapeutic approaches against GPCR activity in melanoma. Recent striking advances in our understanding of the structure, function, and regulation of GPCRs will undoubtedly broaden melanoma treatment options in the future.

## 1. Introduction

G-protein-coupled receptors (GPCRs) participate in intercellular communication by receiving extracellular stimuli from the microenvironment. They then amplify and transduce the signal, passing it on to the nucleus, where it triggers an appropriate cellular response. GPCRs can bind to a wide variety of ligands (hormones, proteins, peptides, amino acids, lipids, nucleotides, xenobiotics, etc.) and regulate numerous essential physiological processes during development and in adult life. GPCRs are the largest receptor family in the mammalian genome, with over 800 members [1]. The GPCR family is composed of receptors that share a common structure, consisting of seven transmembrane helices that are associated with a heterotrimeric G-protein. These receptors are known to regulate many essential physiological processes and their aberrant expression or activity can contribute to human diseases, including cancer. GPCRs are among the most common drug targets because they can be activated or blocked by low molecular weight molecules that have a very strong interaction with their receptors. Their importance in drug discovery is demonstrated by the fact that nearly 60% of drugs in the developmental stage and 36% of currently marketed drugs target human GPCRs, representing around 700 molecules [2]. However, only 10 of these molecules are used in cancer therapy and none have yet been approved for melanoma [3].

Melanoma is a skin cancer that arises from melanocytes, the cells responsible for pigmentation. It affects more than 320,000 people worldwide each year, resulting in the death of nearly 60,000 patients [4]. According to the WHO’s International Agency for Research on Cancer, the incidence is expected to continue to rise in the coming years, reaching half a million patients by 2040. Mortality estimates are also on the rise, with 97,000 deaths estimated for 2040, despite the development of new therapies in the second half of the 2010s [4]. These treatments are based on two approaches: (1) inhibition of the MAPK/ERK pathway in melanoma using a combination of activated BRAF and MEK inhibitors—targeted therapy (TT)—and (2) inhibition of immune cell exhaustion using the checkpoint inhibitors iCTL4 and iPD1—immunotherapy (IT)—[5]. Although these treatments have significantly increased patient survival, half of treated patients relapse [6]. Furthermore, despite significant recent progress in both targeted therapies and immunotherapies for treating advanced-stage disease, the long-term prognosis for patients with cutaneous melanoma is still poor. Therefore, an effective, reliable cure for melanoma undoubtedly requires further therapeutic innovation. Moreover, 40–60% patients are not responsive to current treatments.

GPCRs play crucial roles in various physiological processes, including neurotransmission, cardiac and sensory function, immune responses, and regulation of the pigmentary system. Pigmentation phenotypes have been observed with eight GPCRs (Table 1).

Aside from its cosmetic role, pigmentation is a natural sunscreen that potently absorbs ultraviolet radiation (UVR) and is among the most important factors that determine UV sensitivity and melanoma risk. The melanin pigments responsible for the color of the skin and hair are synthesized within the melanosomes of melanocytes. In the epidermis, melanosomes generated by melanocytes are transferred to keratinocytes to allow homogenous pigmentation and protection of the entire skin against UVR. In the skin, melanocytes, keratinocytes, and dermal fibroblasts communicate with each other via secreted factors and cell-to-cell contact. The crosstalk of the various signaling pathways between these cells constitutes a complex network that controls pigmentation and melanocyte homeostasis. Genetics and in vitro studies have identified loci that regulate pigmentation, and among them, certain key regulators belong to the GPCR family. For example, the identification of the mouse extension locus (extension, recessive yellow, or Mc1r) associated with cloning of the melanocortin 1 receptor (*MC1R*) gene in human melanocytes identified this GPCR as the primary regulator of pigment synthesis. Apart from its effects on melanin production and consequently, UVR protection, the MC1R has functions extending beyond pigmentation, which explain how MC1R activity is directly involved in multiple aspects of melanomagenesis. 

In contrast to epidermal melanocytes, which have a long-life span and low proliferative capacity during adult life, the precursor cells of melanocytes, called melanoblasts, proliferate and actively migrate during embryonic development to colonize the entire skin. Many signaling molecules/ligands are required during all stages of melanocyte development. These ligands educate neural crest cells (NCCs) to specify the melanocytic fate and instruct melanoblasts to proliferate, migrate, survive, and home (towards their final destination) prior to terminal differentiation into pigmented melanocytes. Approximately 100 genes have been shown to be specifically involved in melanocyte development. Among these genes, two GPCRs play a key role: the EDNRB (endothelin receptor type B, with its ligand endothelin 3) and FZL (frizzled receptor, with its ligands WNT1/3A and one of its mediators, β-catenin). Of note, the molecular and cellular mechanisms involved in the proliferation and migration of melanoblasts during development and those of melanoma cells during tumor progression are often closely related. Therefore, it is not surprising to find that key regulators of melanocyte development are also important players in melanomagenesis. The objective of this review is to provide an update on the GPCRs that have an important, well-identified role in melanomagenesis and to discuss the therapeutic strategies that have been used to target these GPCRs.

## 2. Impact of GPCRs on Melanoma Initiation and Progression

### 2.1. Melanomagenesis

During the multi-step process of melanomagenesis, skin melanocytes are transformed into melanoma. Briefly, the first steps consist of the benign proliferation of melanocytes to form a nevus, in which the melanocytes are grouped together and lose their characteristic contacts with keratinocytes. The melanocytes in the nevus eventually stop proliferating and become senescent. As melanomagenesis continues, nevus melanocytes are able to bypass senescence and enter the radial growth phase (RGP), where they typically superficially proliferate toward the basement membrane of the epidermis. These primary steps can be defined as “melanoma initiation”. Then, during the vertical growth phase (VGP), melanoma cells continue to actively proliferate and acquire migratory and invasive properties, allowing them to cross the basement membrane and invade the dermis. The cells eventually acquire metastatic characteristics as they enter the bloodstream and/or lymphatic vessels and eventually colonize various tissues and organs. These latter stages can be considered as the “progression” of the disease. Melanomagenesis is associated with changes in many cellular processes, such as proliferation, immortalization, pseudo-epithelial-mesenchymal transition, migration, and invasion. Cutaneous melanomas are molecularly classified into four groups based on their mutations: BRAF, RAS, NF1, and the “triple wildtype”. All four genomic subtypes of cutaneous melanomas are associated with aberrant activation of the MAPK and/or PI3K/AKT pathway that supports tumor cell growth, proliferation, survival, and anti-apoptotic signals.

The only effective way to demonstrate the causal role of a gene in tumor initiation is to use animal models, whose use allows a better understanding of tumor progression in a physiological context. Indeed, it is very difficult, if not impossible, to reproduce the in vivo cell-cell organization and microenvironment in vitro. However, simple or complex in vitro models are very useful for deciphering the involvement of key genetic elements in various cellular processes, but cannot be used to determine the causal role of a gene in tumorigenesis because (i) established melanoma cell lines are often derived from metastases in which the cells have already undergone a complete transformation process, (ii) the cells are grown on plastic, without their microenvironment, and (iii) colonization of distant organs is very difficult or impossible to assess. Various animal models allow evaluation of the proliferation and bypass of senescence (initiation) and invasion and metastasis (progression). Several animal models have been used to better understand melanomagenesis, including mouse, dog, pig, horse, chicken, and zebrafish. In this review, we focus on the currently best studied mouse models for melanomagenesis.

### 2.2. GPCRs in Melanoma

GPCRs regulate many key biological functions, such as cell differentiation, proliferation, migration, and metabolic activity. Thus, it is not surprising that they play a role in tumorigenesis, including melanomagenesis [15]. There are four main mechanisms by which GPCRs can drive tumorigenesis: (i) excess ligand availability, (ii) excess GPCR expression, (iii) activating mutations in GPCRs, and (iv) activating mutations in Gα proteins. 

The role of certain GPCRs during melanomagenesis has been studied using natural (or chemically induced) mouse mutants of genes of interest or novel engineered gain- and loss-of-function mutants. A list of mouse mutants with pigmentation phenotypes is available and regularly updated (http://www.ifpcs.org/colorgenes/ accessed on 2 January 2022) [16]. In the case of genetically modified mutants, targeting of the melanocytic lineage is performed using the tyrosinase (Tyr), tyrosinase related protein 1 (Tyrp1), dopachrome tautomerase (Dct), or microphthalmia-associated transcription factor (Mitf) promoters/enhancers in the transgenic constructs [17]. Cre recombinase is used to generate conditional mutants, which are required when genes are essential in other lineages and/or during development: Tyr::Cre, Tyr::CreER^T2-Lar^, and Tyr::CreER^T2-Bos^ [18,19,20]. 

The known GPCRs involved in melanomagenesis are presented in the following section. The role of certain GPCRs in melanocyte transformation is predictable, given their key function in melanocyte development and homeostasis (ET/EDNRB, MSH/MC1R, WNT/FZD), whereas the involvement of other GPCRs in melanomagenesis was less expected, such as GRM1, GRM3, GRM5, PAR1, CXCR4, CCR7, CCR10 and GPER1 (Table 2).

#### 2.2.1. Endothelin Receptor Type B (EDNRB)

The endothelin (ET) system consists of two class A G-protein-coupled receptors, endothelin receptors type A and B (EDNRA and EDNRB, respectively) and their three similar peptide ligands, endothelin-1, -2, and -3 (ET1, 2, 3). The EDNRB is the predominant receptor expressed by melanocytes/melanomas and binds all ETs with the same affinity. *Edn3* and *Ednrb* were first found to play a major role during the development of melanocytes from NCCs using genetic knockout mouse models and then, by analogy, to the classic mouse mutants, piebald and lethal spotting [8,36]. Indeed, *Ednrb* and *Edn3* genetically engineered mice (GEM) are both allelic to the spontaneous mouse mutations that occur at the piebald and lethal spotting loci. Recessive mutations at either of these loci give rise to similar phenotypes consisting of differing degrees of hypopigmentation and aganglionic megacolon, due to the absence of enteric ganglia, which have the same neural crest embryonic origin as melanocytes. Reciprocally, increased expression of *Edn3* in the epidermis leads to increased numbers of melanocytes and hyperpigmentation [37]. Moreover, neonatal UV-irradiation of these mice overexpressing Edn3 in the epidermis leads to melanoma formation [38]. Germline *Ednrb* deletion does not lead to tumorigenesis but to the absence of melanocytes, mainly in the dermis. The role of *Ednrb* in melanomagenesis has been evaluated in the context of oncogenic GNAQ^Q209L^ signaling (see next paragraph). The expression of GNAQ^Q209L^ (encoding for Gα_q_) is not sufficient to replace EDNRB signaling during embryonic development, suggesting that Gα_q_ may not be the only G-protein activated downstream of EDNRB (or other signaling pathways). Using a conditional knockout approach, GNAQ^Q209L^–induced melanomagenesis is inhibited in the absence of *Ednrb,* including lung metastases (*Mitf-cre*/+; *Rosa-fs-GNAQ^Q209L/+^*; *Ednrb^F/F^* background) [21]. Intriguingly, germline haploinsufficiency for *Ednrb* has the opposite effect in the RET mouse melanoma model (*Metallothionein-1/RFP-RET*; *Ednrb*^+/−^ mice), in which it accelerates tumorigenesis, with an increase in lung metastases [39]. These two mouse models are of interest and clearly show that *Ednrb* expression affects melanomagenesis. However, they have the disadvantage of representing an uncommon oncogenic situation in human cutaneous melanoma, with haploinsufficiency or the lack of *Ednrb* combined with a GNAQ^Q209L^ driver mutation or increased RET signaling. 

Interest in EDNRB in melanoma stems primarily from early observations in humans showing that EDNRB expression was positively associated with cutaneous melanoma progression; EDNRB mRNA and protein levels were found to increase from common nevi to dysplastic nevi and from primary to metastatic melanoma [40]. Consistent with this observation, in vitro experiments showed that ET promoted melanoma cell proliferation, migration, and invasion and that EDNRB inhibitors reduced melanoma cell growth and survival in culture and xenografts [41,42,43]. However, overexpression of EdnrB alone or combined with driver mutations was not performed in mice to genetically address its role in tumorigenesis in physiological situations.

Another aspect of ET signaling in melanoma is its activity in DNA repair, which has a role in reducing the genotoxic effect of UVR [22]. Indeed, ET signaling increases intracellular Ca^++^ mobilization and downstream activation of the stress-induced MAP kinases JNK and p38 in UV-irradiated human melanocytes. This activation in turn enhances the repair of cyclobutane pyrimidines (CPDs), the major form of DNA photoproducts [44]. Finally, a recent study has suggested that ET signaling has a multifunctional role in melanoma, acting on both tumorigenic and stromal cells, where it mediates immunosuppression by increasing Treg proliferation [45]. Thus, although the role of EDNRB signaling is relatively well understood during melanocyte development, its role in malignant transformation is much less clear, as it acts in multiple signaling pathways and is context dependent. As such, it is not surprising that therapies that target EDNRB have thus far not been very successful. Small-molecule inhibitors of EDNRB, A-192621 and BQ788, were shown to inhibit the growth and survival of melanoma cells in culture and in xenografts [42,43,46]. However, the dual EDNRA/EDNRB antagonist, Bosentan, was tested in phase II clinical trials and failed to produce a robust response in cutaneous melanoma patients, neither alone nor in combination with dacarbazine [47,48]. Similarly, A-192621 treatment of mice expressing Ednrb in the context of oncogenic GNAQ^Q209L^ showed no effect on tumorigenesis, whereas haploinsufficiency for Ednrb reduced it. Targeting EDNRB with an antibody-drug conjugate (DEDN6526A) is currently being tested in phase I [49]. It would be of great interest to generate a mouse model that reflects the human EdnRB situation in cutaneous melanoma: overexpression of EDNRB (human and mouse) in melanocytes/melanoma combined with oncogenic BRAF^V600E^ or NRAS^Q61K/R^, the major driver mutations in human cutaneous melanoma. Such models would allow a better understanding of the effect of *Ednrb* overexpression on melanomagenesis, the study of its downstream signaling, and the testing of its inhibitors in a human relevant preclinical mouse model before clinical trials. 

#### 2.2.2. The Melanocortin Receptor (MC1R)

Melanocytes express a receptor (**MC1R**) that controls melanogenesis. The MC1R belongs to a small subfamily of GPCRs, classified into five subtypes (MCR1-5) that contribute to important physiological processes. MC1R is the only melanocortin receptor expressed in melanocytes. MC1R is a class A receptor and is coupled to Gs protein. MC1R binds to the pro-opiomelanocortin-derived peptide α-melanocyte-stimulating hormone (α-MSH), resulting in the activation of downstream signaling cascades in a cAMP-PKA-dependent manner [50]. Upon UV exposure, α-MSH is released by keratinocytes, leading to stimulation of the MC1R at the melanocyte membrane, the activation of protein kinase A (PKA), and ultimately, to increased cAMP levels. Other pathways independent of UV exposure can lead to increased α-MSH production and hyperpigmentation, as observed in the Dopamine receptor D2 knockout (DRD2tm1Ebo) [7]. An important target of cAMP is the transcription factor CREB (CAMP-responsive element-binding protein), which becomes phosphorylated and then activates the promoter of *MITF*, which in turn up-regulates the transcription of the melanogenesis enzyme genes *TYR*, *TYRP1*, and *DCT*, as well as those regulating other cellular processes, including proliferation, invasion and metabolism [51,52]. In addition, binding of neurofibromin 1 (NF1) to MC1R regulates intracellular signaling pathways involved in pigmentation [53]. MC1R is the product of the gene located at the *extension* locus and stimulates the synthesis of the pigment, eumelanin (black, brown). The loss-of-function mutation in this locus, *recessive yellow* (*e*/*e*), results in the production of pheomelanin (yellow, red) instead of eumelanin [13,54]. An MC1R antagonist is the agouti signaling protein (ASP). Mutations in the mouse Agouti gene that cause increased and ectopic expression of ASIP (*viable yellow*, *Avy*) result in yellow coat color, similar to the phenotype of *e*/*e* mice, as well as obesity due to ASIP binding to the MC4R. In humans, more than 200 MC1R variants have been identified and high numbers of natural *MC1R* variants are strongly associated with pigmentary phenotypes, providing evidence that the MC1R is the main determinant of human pigmentation and central to eu- and pheo-melanin regulation [50]. Similar to mice, MC1R variants in humans can result in the reduction of receptor activity and a shift in melanin synthesis from eumelanin to pheomelanin. MC1R is inactivated in people with red hair, due to polymorphism(s) that make(s) them more susceptible to melanoma than dark-skinned individuals. For example, variants of the gene encoding *MC1R*, mainly R151C, R160W, and D294H, have been shown to be associated with light and poorly pigmented skin [55], whereas the WT form is associated with dark, highly pigmented skin [56,57]. Similarly, patients with mutations in proopiomelanocortin (*POMC*) genes—encoding the precursor of αMSH—have red hair [58] These variants decrease the sensitivity of the receptor and binding of the hormone α-MSH, produced by keratinocytes in response to UVR. Epidemiology studies have strongly established that the MC1R functions as a melanoma predisposition gene. However, it is still not clear whether this is due to the lack of eumelanin, to photoprotective and antioxidant activities, to the expression of pheomelanin, which is known to amplify UVA-induced reactive oxygen species (ROS), or to other functions not related to pigmentation. Indeed, pheomelanin was shown to promote melanomagenesis via the induction of oxidative DNA damage, without exposure to any carcinogens, such as UVR, in mice harboring the activating Braf^V600E^ mutation combined with MC1R^e/e^ [23]. Thus, loss of function of MC1R promotes initiation in a UV-independent manner, demonstrating its tumor suppressor activity and a key role in the initiation of melanoma.

Apart from its central role in pigment switching, it is now recognized that MC1R has non-pigmentary roles in antioxidant defenses and DNA-repair mechanisms [59,60,61]. The cAMP pathway enhances melanocyte nucleotide excision repair (NER) activity, which operates by a « cut and patch » mechanism, to remove UV lesions. Activation of the MC1R by α-MSH binding results in phosphorylation of the DNA damage sensors ataxia telangiectasia mutated (ATM) and Rad3 related (ATR), as well as in the recruitment of the xeroderma pigmentosum complementing proteins XPC (Group C) and XPA (Group A) [22]. Consistent with MC1R promoting DNA damage repair, impairment of the NER pathway in subjects carrying an *MC1R* loss-of function mutation has been observed. Additional non-pigmentation-related effects of MC1R can be attributed to the activation of MITF expression, which controls genes involved in DNA damage repair, chromosome stability, and centromere integrity [62]. 

Therapies involving defective MC1R signaling aim to restore its activities. Mouse mutants of MC1R have been characterized for years and can be used to evaluate therapies for better protection against UVR. Topical application of the cAMP permeable-inducer forskolin onto mice harboring loss-of-function mutations or haploinsufficiency of Mc1r^e/e^ stimulated eumelanogenesis and induced UV-resistance [24,63]. These studies confirmed epidemiological studies suggesting that MC1R haploinsufficiency increases mutagenic susceptibility to UVR and melanoma risk. Another therapeutic approach is to use MC1R agonists to increase pigmentation, antioxidant defense, and DNA repair. The best-known analogue is NDP-MSH, which is 100 times more potent than α-MSH and is currently used to treat photosensitivity diseases, such as erythropoietic protoporphyria (EPP). A very promising analog is the tripeptide (LK-514), which is >10^5^ times more selective for MC1R than other melanocortin receptors [64]. The challenge of MC1R-based therapies is to use an analog that is highly specific to MC1R to prevent toxic effects due to the activation of other receptors and to avoid targeting expression of MC1R in non-melanocytes.

#### 2.2.3. The Wnt/Frizzled Receptor

The Wnt (fusion of the words wingless and integrated) pathway is one of the most important signaling pathways during embryonic development and adult homeostasis and its deregulation has often been linked to cancer. Wnt proteins activate at least three different intracellular signaling pathways: the Wnt/β-catenin (or canonical), Wnt/Ca^2+^, and Wnt/planar cell polarity pathways [65,66]. The type of Wnt protein secreted determines which of these three signaling cascades is activated. The Wnt family contains at least 19 secreted cysteine-rich glycoproteins in humans [67,68]. Wnt proteins bind to target cells via two families of receptors: the seven transmembrane receptors Frizzled (**Fzd**) and LDL-receptor-related proteins. The Frizzled (FZDs) receptors are comprised of ten members (FZD1–FZD10), most of which are coupled to the β-catenin (bcat) canonical signaling pathway.

The Wnt/β-catenin pathway is essential for melanocyte development from NCCs [69]. The ligands Wnt1 and Wnt3a are required for the specification, expansion, and differentiation of melanoblasts from NCCs [70,71]. β-catenin itself has been directly implicated in melanoblast determination in several models, with varying effects depending on the temporality of its activation [72,73]. In mice, loss of bcat from pre-migratory NCCs (Wnt1::Cre; Ctnnb1ex2-6^F/F^) or melanoblasts (Tyr::Cre; Ctnnb1ex2-6^F/F^) induces the disappearance of melanoblasts [73,74]. The expression of a stabilized form of bcat (Tyr::bcat-mut-nls-egfp) leads to mice with a ventral white coat area associated with a defect in melanoblast migration [75]. Various β-catenin targets have been shown to be involved in cell proliferation and include the ubiquitous genes Myc and CyclinD1 and the melanocyte-specific gene Mitf-M.

Only one member of the Frizzled family, FZD4, has been implicated in pigmentation. FZD4 knockout (Fzd4^tm1Nat^) induced a depigmentation of coat color in addition to multiple defect in the nervous system [9]. FZD receptors are frequently overexpressed in tumor tissues relative to normal tissues and are potentially associated with a poor prognosis. No FZD overexpression has been directly linked to melanoma. Nevertheless, the FZD7 receptor was found to be upregulated in metastatis-derivative melanoma cell lines compared to the parental A375P cell line, with its expression associated with amoeboid invasion [26]. FZD7 knock-down reduces tumor growth after subcutaneous injection of the WM1361 melanoma cell line as well as metastasis formation after tail-vein injection of several melanoma cells in NSG mice [25,26]. OMP-18R5, a monoclonal antibody targeting several FZD including FZD7, is able to block tumor growth in xenograft mouse models for multiple cancers, but was not evaluated for melanoma [76].

Although little evidence implicates FZD receptors in melanomagenesis, studies of its associated signaling with β-catenin clearly highlight a central role of this pathway in melanoma. Melanoma was one of the first cancers in which *CTNNB1* mutations were identified. In mouse models, the activation of WNT/β-catenin signaling participates in the initiation of melanomagenesis but is not, alone, sufficient for initiation. Expression of a stabilized mutated bcat in melanocytes, along with a mutated human NRAS oncogene, constitutively activating the MAPK pathway (*Tyr::NRAS^Q61K^*/°; *Tyr::bcat-mut/°*) led to accelerated onset and increased the number of melanomas in a mouse model [77]. This property has been linked to the increased immortalization of melanocytes in vitro by the repression of p16 expression. In a Braf^V600E^; Pten^-/-^; bcat-null-KO (*Tyr::CreER^T2-Bos^*; *Braf^CA^*; *Pten^F/F^*; *Ctnnb1ex2-6^F/F^*) mouse model, in which β-catenin is inactivated, the occurrence of melanoma is strongly delayed relative to that in a Braf^V600E^; Pten^-/-^ model [78]. In a Braf^V600E^; Pten^-/-^; bcat (*Tyr::CreER^T2-Bos^*; *Braf^CA^; Pten^F/F^*; *Ctnnb1ex3^F/F^*) mouse model, in which bcat is activated, the occurrence of melanoma is accelerated [78]. In mouse models of Braf^V600E^, Pten^-/-^, and NRAS^Q61R^ melanoma, bcat activation increases the number of lung metastases, whereas bcat inactivation decreases the number of lymph-node and lung metastases [75,78]. In conclusion, the activation of β-catenin increases both the initiation and progression of melanoma in mouse models. Of note, in humans, several studies have linked the Wnt/β-catenin pathway to the antitumor immune response in melanoma [79,80,81].

#### 2.2.4. Glutamate Receptors (GRM1, GRM3 and GRM5)

Glutamate is the most abundant excitatory neurotransmitter in the human central nervous system, where it plays a critical role in intercellular communication. Glutamate receptors are also expressed in tissues outside of the nervous system and are involved in the modulation of various normal and pathological processes. The glutamate receptor family is divided into two major groups: ionotropic glutamate receptors (iGluRs) and metabotropic glutamate receptors (Grms). The Grm1s belong to the class C family of GPCRs, characterized by a large, globular, extracellular ligand-binding domain. The Grm1 family consists of eight members (Grm1-8), which are organized by sequence homology, signaling effectors, and general localization. Group I Grms, consisting of **Grm1** and Grm5, are multi-coupling receptors that can signal through both the Gα_q_ and Gα_i/o_ pathways. Group II Grms consist of Grm2 and Grm3 and couple to the Gα_i/o_ pathway. Group III Grms consist of Grm4, Grm6, Grm7, and Grm8 and couple to Gα_i/o_ signaling pathways. Three members of the metabotropic glutamate receptors (mGluR1, mGluR5, mGluR3) have been clearly identified as regulators of melanomagenesis (see [82] for review).

The involvement of metabotropic glutamate receptors in melanomagenesis was initially revealed by chance in a complex study using insertional mutagenesis, leading to aberrant expression of Grm1. Surprisingly, the mouse developed metastatic melanoma, whereas Grm1 is not detected in normal melanocytes in mice [83]. Confirming this initial observation, transgenic mice containing *Grm1* under the control of the Dct promoter (Dct::Grm1) developed melanoma with 100% penetrance [12]. Initially, no distant organ metastases were observed but disseminated cells were later detected in distant organs, such as the lung and liver [28]. The importance of Grm1 on tumor growth in vivo was supported by the decrease in growth induced by Grm1 knock-down in xenografted cells [84]. A conditional transgenic model using the tetracycline-regulated system to express mGluR1 in adulthood demonstrated that *Grm1* expression is required not only for the initiation of melanoma but also for its progression in vivo [85]. The gene encoding the human receptor (*GRM1*) is altered in melanoma by point mutations, amplification, and/or deletions. In humans, GRM1 expression is not detected in normal melanocytes but it is expressed in 80% of metastatic melanoma or cell lines. Works from the laboratory of S. Chen showed that GRM1 expression results from activation of the MAPK and PI3K/AKT pathways, the main pathways activated in melanoma [82,86,87]. Melanomas expressing GRM1 show elevated levels of glutamate in the tumor microenvironment, contributing to hyperactivation of the receptor and its downstream effectors. The identification of this autocrine loop between GRM1 expression and the secretion of glutamate led to clinical trials to test riluzole, known to reduce glutamate release and thus reduce activation of the receptor. Despite being efficient in mice, no objective responses were observed in humans [88,89]. Recently, the secretion of glutamate by melanoma cells has been associated with an increased glutaminolysis due to aberrant expression of glutaminase. Co-inhibition of glutaminolysis and GRM1 have proven to be efficient in decreasing melanoma xenograft growth in vivo [90]. This co-treatment remains to be tested in the clinic. Furthermore, riluzole monotherapy showed more immune cell infiltrates in stable disease patients than those with progressive disease, suggesting that combining riluzole with immune checkpoint blockade therapy could enhance the efficacy of either agent alone. However, this has not yet been tested.

In contrast to Grm1, Grm5 is normally expressed in both normal melanocytes and melanoma tumors. Transgenic mice overexpressing ***Grm5*** (*Tyrp1-Grm5*) present with multiple melanoma located on the tail, with 100% penetrance and metastases, demonstrating that *Grm5* drives melanoma initiation and progression [30]. Of note, mice with *Grm1* or *Grm5* melanoma both exhibit tumor formation on the hairless skin, including the pinnae and tails, rather than on trunk areas, as observed for BRAF^V600E^ and NRAS^Q61K^ induced melanoma, suggesting that the origin of the transformed cell may not be identical (epidermal vs. hair follicle). No information concerning GRM5 in humans is available.

Exon capture sequencing of 734 GPCRs in malignant melanoma showed that a third glutamate receptor, ***GRM3***, is frequently mutated in human melanoma. The identification of the same mutations (G18E/R, M518I) in multiple individuals suggests that these mutations may be “drivers” of the oncogenic process. GRM3 mutants selectively regulate the phosphorylation of MEK, leading to increased anchorage-independent growth and migration [29]. Mutated GRM3 cells are more sensitive to MEK inhibitors. To date, no transgenic models with GRM3 mutants to study its effect on cellular transformation and sensitivity to MEK inhibitors have been developed. GRM3 mutants may contribute to melanomagenesis through cross talk between the cAMP and MAPK signaling pathways [91]. The misexpression of two other glutamate receptors, *GRM4* and *GRM8*, has been detected in melanoma, but their precise role has not yet been clearly demonstrated [82]. Over 60% of human melanomas express *Grm*, indicating the importance of glutamatergic signals in this type of tumor. It appears that glutamate receptors are not only involved in neuronal signaling and neuronal disorders but also in the transformation of the originating neural crest-derived cells, melanocytes, into melanoma. Note that other NCC derivatives express GRMs. As an example, GRM1 is expressed in chromaffin cells of the adrenal medulla and GRM5 is expressed in the Meckelian cartilage [92,93]. However, no other NCC derivatives transformed by glutamatergic signaling have been reported so far. 

The GPCRs that are cited in the next paragraphs (PAR1, CXCR4, CCR7, CCR10 and GPER1) have been implicated in melanomagenesis. However, their functional role in melanoma initiation and in an immunocompetent environment have not been studied.

#### 2.2.5. PAR1 

The protease-activated receptors (PARs) are a family of GPCRs comprised of four members (PAR1–4) involved in the regulation of various cellular processes, including inflammation and coagulation. Cleavage of **PAR1** (also known as the thrombin receptor) by thrombin activates the receptor and downstream signaling through multiple heterotrimeric G-proteins such as Gα_q_, Gα_i/0_, and Gα_12/13_. In turn, the MAPK and PI3-K signaling pathways are activated, along with phospholipase C-β (PLC-β). Elevated PAR-1 expression during melanoma progression has been suggested to promote key processes that contribute to melanoma metastasis. Targeting PAR-1 reduced tumor growth and the metastases of melanoma cells in xenograft experiments [31]. Overexpression of PAR-1, as well as the continuous activation of thrombin, promotes the upregulation of genes involved in adhesion, invasion, angiogenesis, and metastasis [94]. As PAR-1 signaling affects both melanoma cells and their microenvironment, it was considered to be an attractive therapeutic target for the treatment of melanoma patients. However therapeutic trials were not continued in melanoma due to the activity of PAR1 in coagulation.

#### 2.2.6. Chemokine Receptors (CXCR4, CCR7, CCR10) 

Chemokine receptors belong to the GPCR family and are classified into four groups, CXCR, CCR, XCR, and CX3CR. Each receptor can bind to several chemokines. A variety of chemokine receptors are expressed on the surface of both immune and tumor cells. Expression of **CXCR4**, **CCR7** and **CCR10** on the surface of melanoma cells is associated with a poor prognosis [95]. Aside from their critical role in the immune response, chemokines and their receptors have been studied for their capacity to guide cancer cells to specific organs. Chemokines have chemotactic properties and can attract melanoma cells expressing their corresponding receptors. High concentrations of CXCL12, the ligand of CXCR4, are produced in the lungs and injected CXCR4-expressing B16 melanoma cells are able to efficiently colonize the lung. Such colonization is reduced in the presence of T22, a specific inhibitor of CXCR4 [32]. In addition, a commercially available dermal filler, hyaluronic acid (HA)-based gel, loaded with CXCL12 was able to recruit and trap CXCR4-expressing B16 melanoma cells injected into mice, consequently leading to a reduction in lung metastases [33]. One hundred and thirty-eight (138) clinical trials with CXCR4 inhibitors are being/have been performed, none of which include(d) melanoma. In mice, the overexpression of CCR7 or CCR10 in B16 melanoma cells was shown to increase regional lymph node metastases, which was blocked by neutralizing its ligand, CCL21, using a specific antibody [34,35,95]. However, most of these experiments used mouse B16 melanoma cells, which are not necessarily the best representation of human melanoma. The effect of the various chemokines on the immune response is not discussed here. 

#### 2.2.7. G-Protein-Coupled Estrogen Receptor 1 (GPER1)

There are three estrogen receptors, two nuclear receptors, ERα and ERβ, that act mainly as transcription regulators, and the G-protein-coupled estrogen receptor 1 (**GPER1** = GPER = GPR30), that can induce rapid, non-genomic estrogen signaling [96]. GPER1 coupled with Gα_s_ protein induces the cAMP pathways. Using various mouse models, GPER1 has been shown to play pleiotropic functions particular in the endocrine, immune, cardiovascular and central nervous systems. In these mice, no pigmentary phenotype has been observed. The impact of estrogen in melanomagenesis is still controversial since multiple genetic and environmental factors can greatly influence the development of this cancer and its severity. Nevertheless, Natale at al., propose that repeated pregnancies inhibit the growth of BRAF-driven human melanocytic neoplasia in xenografts and that GPER1 signaling promotes cell differentiation instead of proliferation, inhibiting tumor development [27]. Furthermore, in this study, GPER1 signaling rendered melanoma cells more vulnerable to immunotherapy. These results led to the initiation of a clinical trial targeting GPER1 and anti-PD1 immunotherapies (Phase I/IIA trials: NCT04130516). It should be pointed out that there is an notable controversy concerning the role of GPER1 as the principal mediator of the estrogen response in vivo, because even though nuclear receptors are less expressed in melanoma than GPER1 (Median level for GPER1 = 1.86; Erα = 0.61; Erβ = 0.31Tpm according to TCGA data base), they have a higher affinity for 17β-estradiol (GPER1 = 3–6 nM and ERs = 0.1–1 nM) [97]. It is clear that the role of GPER1 in melanoma needs further investigation, particularly in physiologically relevant mouse melanoma models.

Other GPCRs, namely GPR143, GPR161, SMO, which couple to Gα_q_, Gα_s_ and Gli, respectively, have produced pigmentation phenotypes in mice, but have not been implicated in melanomagenesis [10,11,14].

## 3. GPCR Associated Signaling in Melanoma

The activation of GPCRs leads to the modulation of the activity of cellular signaling pathways, marked by the production of second messengers. The first element of such signaling is the heterotrimeric G-proteins to which these receptors are coupled through their intracellular domain. These G-proteins consist of three subunits, Gα, Gβ, and Gγ [98]. The C-terminal subunit of the GPCR is responsible for the selectivity of receptor/G-protein binding [98,99]. The G subunit is also responsible for the selectivity of downstream signaling pathways [100]. There are a total of 17 Gα subunits grouped into four subfamilies: Gα_s_, Gα_i/0_, Gα_q/11_, and Gα_12/13_.

GPCRs generally couple to a specific G-protein but may interact with several different G-proteins [56,101]. Coupling appears to be cell-dependent. Thus, a careful analysis of the downstream signaling is required for each cell type. Binding of a ligand to its GPCR causes a conformational change in the GPCR that is transmitted to the Gα subunit, which exchanges its GDP to a GTP molecule. The binding of GTP induces the dissociation of Gα from the Gβ-Gγ subunits of the receptor. This dimer then modulates the activity of other intracellular proteins [98]. The GαGTP and Gβ/Gγ complexes then generate different intracellular signals that included cAMP, inositol-1,4,5 trisphosphate (IP3), diacylglycerol (DAG), and Rho proteins (Figure 1).

### 3.1. Signaling via cAMP

The induction of MC1R by its ligand activates the **cAMP** pathway, whereas the induction of GRM1, GRM3 and GRM5, CXCR4, CCR4, CCR7, and CCR10 represses this pathway. cAMP was the first second messenger to be discovered and regulates many downstream cellular processes [102,103]. Two classes of G subunits modulate intracellular cAMP levels, Gα_s_ and Gα_i/0_, with diametrically opposite effects [100,104]. The enzyme responsible for cAMP production, adenylate cyclase (ADCY), is a membrane-associated enzyme that converts ATP to cAMP [104,105]. The difference between Gα_s_ and Gα_i/0_ is due to the difference in the binding domain on the adenylyl cyclase: members of the Gα_s_ family bind to the C2 intracellular domain of the adenylyl cyclase, which then activates its enzymatic activity, ultimately leading to an increase in intracellular cAMP levels [106]. Conversely, members of the Gα_i/0_ family bind to the C1 intracellular domain of the adenylyl cyclase, thereby inhibiting its activity. 

The heterotrimeric G-protein subfamily Gα_s_ is composed of three members: Gα_s_ and Gα_sxl_, two splice variants of the *GNAS* gene, and Gα_olf_, encoded by the *GNAL* gene [100]. Only Gα_s_ is expressed in melanocytes and melanoma [107] and is activated by an associated receptor, such as the MC1R [108]. Gain-of-function mutations of *GNAS* are clustered at amino acids R201 and Q227, located in the GTPase activity pocket. These mutations induce the loss of intrinsic GTPase activity and maintain the Gα_s_ protein in an activated state. Mutations of GNAS are frequently found in various tumors of the pancreas, kidney, and stomach, but are more anecdotal in melanoma, affecting less than 1% of cases [109,110]. More strikingly, the *GNAS* T393C SNP polymorphism is associated with tumor progression in metastatic melanoma [111], as well as in other cancers, such as colorectal or bladder cancer [112,113]. How this polymorphism affects the activity of Gα_s_ and the oncogenic process has not been evaluated. However, this observation suggests that Gα_s_ inactivation favors melanoma progression.

The Gα_i/0_ heterotrimeric G-protein subfamily is composed of eight members: Gα_i1_, Gα_i2_, Gα_i3_, Gα_0_, Gα_z_, Gα_t-r_, Gα_t-c_, and Gα_gust_. However, only **Gα_i1_**, **Gα_i2_**, and **Gα_i3_** are expressed in melanocytes and melanoma, and are encoded by *GNAI1-3,* respectively [100,110,114,115]. Gain-of-function mutations of *GNAI2* are clustered at amino acids R179 and T182 and lead to constitutive activation of the Gα_i2_ subunit by increasing its GTP binding capacity [116,117]. *GNAI2* is mostly involved in cell injury and inflammatory responses but these activating mutations can lead to tumors, depending on the cellular context, due to increased MAPK activity. Gα_i2_ R179 and T182 mutations are found in 1.4% of melanoma patients [110] but their impact on melanomagenesis has not yet been evaluated.

Gα_s_ and Gα_i/0_ directly regulate adenylate cyclase. There are ten (10) enzymes encoded by ten (10) different *ADCY1-10* genes [118,119]. All are membrane bound via their two series of six transmembrane helices (TM1 and TM2), followed by a cytoplasmic domain (C1 and C2, respectively). Only ADCY10, which is soluble, differs from the others, as its activation is GPCR independent, being activated by bicarbonate and calcium [120]. Most adenylate cyclases are expressed in melanoma, with the exception of ADCY5 and ADCY8 [110,115]. The expression of ADCY10 is unclear because, although teams have found the protein by IHC, databases suggest that the mRNA is absent [110,121]. It has been shown that metastatic melanomas express more ADCY1 mRNA than primary melanomas and that a high level of ADCY1 expression correlates with a poorer prognosis [122]. Consistent with this observation, the silencing of ADCY1 in vitro in mucosal melanoma cell lines decreases the ability of cells to form clones in a colony-formation assay, as well as their migratory and invasive capacity [123]. In xenograft experiments, decreased ADCY1 expression decreased subcutaneous cell growth as well as colonization of the lung after the injection of melanoma cells into the tail vein of NOD/SCID mice [123]. More generally, stimulation of adenylate cyclase activity by forskolin promotes tumor growth in the *Braf^CA^*/*Pten*^−/−^ melanoma mouse model, whereas its pharmacological inhibition by SQ22536 leads to a decrease in tumor growth in a MAPK pathway-independent manner [124,125]. However, treatment of human primary and metastatic melanoma cell lines with SQ22536, even at high concentrations, does not alter cell survival. This implies that the targeting of transmembrane adenylate cyclase is not a feasible therapeutic strategy on its own. Adenylate cyclase activity is also involved in resistance to MAPK inhibitors. Indeed, treating BRAF^V600E^ melanoma cells with forskolin increases ADCY9 expression and cAMP synthesis, leading to greater resistance to MAPK inhibition [126].

The intracellular concentration of cAMP is negatively regulated by phosphodiesterases (PDEs), which hydrolyze cAMP to AMP, thus controlling the amplitude and duration of the signal. There are 11 families of PDEs (PDE1-11) encoded by a total of 21 different genes [127]. Specific PDE isoforms are located in different subcellular compartments, where they regulate cAMP levels. Indeed, cAMP does not freely diffuse across the cell but is rather produced in subcellular compartments. This feature has important consequences, allowing only appropriate targets to be activated in microdomains [128].

In melanocytes, phosphodiesterase 4 (PD4E), more specifically its variant PD4ED3, is a direct target of MC1R-cAMP signaling, constituting a negative feedback mechanism [129]. Blocking PDE4D3 activity in conjunction with forskolin treatment can efficiently restore cAMP levels and pigmentation in MC1R^e/e^ mice. In melanoma, the expression of numerous PDEs has been reported and their effect was initially shown on cell proliferation [130,131]. Their specific functions in BRAF or NRAS-mutated melanoma highlight the connection between the cAMP and MAPK pathways (see below). The overexpression of PDE4 and, therefore, the inhibition of cAMP signaling is critical for MAPK activation by oncogenic RAS in melanoma [131,132]. In BRAF-mutated melanoma, the inhibition of PDE4 activity by pharmacological inhibitors or RNA interference decreases melanoma cell invasion by interacting with the focal adhesion kinase FAK [133]. Overall, 3.5% of solid tumors (including melanoma) have homozygous microdeletions of PDE4D associated with increased expression and a tumor-promoting effect [130]. PDE4D expression is elevated in advanced melanoma and negatively associated with survival. More generally, inhibiting cAMP signaling through the expression of PDEs (PDE1, PDE2, PDE4, and PDE8) is associated with the oncogenic progression in melanoma [128]. Whether PDE inhibitors can prevent proliferation, invasion, and/or migration in melanoma needs to be evaluated in the future. 

The first and main target of cAMP is protein kinase A (PKA) [100,134]. PKA is a serine/threonine kinase composed of four subunits: two regulatory and two catalytic. There are four isoforms for both the regulatory (**RI**α, **RI**β, **RII**α, and **RII**β) and catalytic subunits (**Cα**, **Cβ**, **Cγ**, and **PRKX**), with each isoform showing individual localization and specificity. In humans, all are expressed, but only Cγ is expressed in melanocytes and melanomas [135]. Theoretically, since RIs cannot interact with RIIs, a maximum of 36 combinations can be generated between regulatory and catalytic subunits. At this point, specific combinations have not been characterized in melanocytes or melanomas.

Binding of cAMP to the regulatory subunits induces their dissociation from the catalytic units, which become active and phosphorylate downstream targets [136]. More than 70% of patients with familial Carney complex, an autosomal dominant skin condition associated with spotty pigmentation [137], carry three mutations associated with pathogenic features (82C>T, 491_492 delTG, c.709-2_709-7 delATTTTT) in the *PRKAR1A* gene, which encodes the RIα subunit [138]. This mutation induces a dominant-negative action of the regulatory subunit and in consequence a constitutive activation of the catalytic subunit of PKA. Inactivating mutations of PRKAR1A lead to constitutive activation of the cAMP-PKA pathway through the loss of regulation of the catalytic subunits of PKA. In melanoma, mutations are found in 1.4% of cases, as well as loss of heterozygosity in 11.8% of patients [110]. Furthermore, loss of function of *PRKAR1A* is found in epithelioid pigmented melanocytomas, a rare intermediate/borderline form of melanoma [139]. By comparing PKA activity in primary and metastatic melanoma cells, Beebe and colleagues suggested that PKA activity is higher in melanoma metastases [140]. The pharmacological inhibition of PKA induces the growth and invasion of melanoma cells [141].

A large number of cytosolic and nuclear proteins have been identified as substrates for PKA [102]. Importantly, PKA is located at the crossroads between cAMP and MAPK/ERK. Constitutive activation of cAMP leads to the phosphorylation and inactivation of CRAF by PKA in melanoma [131]. CRAF is important in maintaining activation of the MAPK pathway in RAS-mutated cancers because ERK1/2 has a negative feedback action through the phosphorylation of BRAF, which causes its inhibition [131,132,142]. As a consequence, activation of MAPK pathways through CRAF requires that the cAMP pathway in melanoma cells be inactivated to release cAMP-mediated inhibition of CRAF. PKA can also directly phosphorylate BRAF on serine 365, dissociating the RAS/BRAF/KSR complex and thus activating BRAF [143]. The catalytic Cα subunit renders BRAF^V600E^ melanoma cells resistant to MAPK inhibitors [126]. The CRTC3 protein, a co-activator of CREB that is phosphorylated and activated by PKA and ERK [144], lies at the interface between signaling through the cAMP and MAPK pathways. A knockout mouse model for Crct3 showed graying of the coat due to defects in melanocyte maturation [144]. Mutations in *CRCT3* have been identified in 23% of human melanomas, most leading to an increase in its expression and activity and reduced patient survival [144]. Thus, CRT3 inhibition could be beneficial for such patients. 

The regulation of transcription by PKA is mainly achieved by the phosphorylation of CREB. CREB phosphorylation leads to dimerization of this transcription factor and its subsequent binding to cAMP response elements (CRE) in target genes and its interaction with transcription co-activators, such as CREB-binding protein (CBP) and p300. CRE binding sites are located in the promoter regions of many genes, including the master melanocyte regulator MITF [51,145]. MITF regulates numerous major cellular processes essential for melanogenesis and melanomagenesis, including pigmentation, growth, survival, migration, and invasion [52]. In melanoma, CREB overexpression is associated with transition from the radial to vertical growth phase [146]. 

The inhibition of CREB in melanoma cell lines was shown to decrease metastasis formation after injection into the tail veins of mice [147,148]. This loss of metastatic potential can be explained, at least in part, by the loss of the expression of the metalloproteinase MMP2 and the adhesion molecule MCAM/MUC18. Surprisingly, CREB, which generally acts as a transactivator, negatively regulates the transcription factor AP2α and the gene encoding cellular communication network factor 1 (CCN1/CYR61). In early publications, these two genes were considered to be tumor suppressor genes in melanoma but more recent studies have proposed that AP2α and CCN1 facilitate melanoma progression [149,150,151]. The RNA-editing enzyme adenosine deaminase acting on RNA1 (ADAR1) has been recently identified as a new target of CREB. Silencing ADAR1 enhances the invasiveness of melanoma cells [152]. CREB has been associated with resistance to MAPK inhibitors. Phospho-CREB is restored in relapsing melanomas previously treated by MAPK inhibitors, possibly by the up-regulation of adipocyte enhancer-binding protein 1, AEBP1 [126,153].

The second major target of cAMP is the cAMP-activated exchange protein (**EPAC**) [154]. EPAC is a guanine nucleotide exchange factor (GEF) for small GTPases, e.g., RAP1 (Ras-related protein 1). Activation of RAP1 occurs through the exchange of GDP for GTP [102,154]. The consequences of EPAC activation on the growth of melanoma are still unclear and reports are conflicting. Indeed, pharmacological activation of EPAC using an EPAC-specific cAMP analog increases the growth of HMG cells [155] but has no effect on PMP melanoma [141]. The use of shRNA targeting Rap1 increases the growth and survival of cells derived from primary but not metastatic melanomas [124]. The current hypothesis is that the EPAC-Rap1 pathway is anti-proliferative in metastatic melanoma and pro-proliferative in primary melanoma [156].

The last major effectors of cAMP are the cAMP-dependent ion channels: the cyclic nucleotide-gated ion channel (CNG) and the hyperpolarization-activated cyclic nucleotide-gated channel (HCN) [154,157]. These channels are relatively nonselective cation channels and have not yet been studied in melanoma.

### 3.2. Signaling via Inositol Triphosphate and Diacylglycerol

The induction of EDNRB and PAR1 by its ligand activates the **IP3/DAG** pathway. It has to be noted that GRM1 and GRM5 induce the IP3/DAG pathway, but repress the cAMP pathway. Cellular levels of IP3 and DAG are highly regulated by the stimulation of GPCRs of the **Gα_q/11_** class. This class contains four members: Gα_q_, Gα_11_, Gα_14_, and Gα_16_ [100]. All can be expressed in melanocytes and melanoma but only Gα_q_ and Gα_11_ are highly expressed in these cells [110,115]. Activation of the receptor induces the exchange of GDP bound to the alpha subunit for GTP, resulting in activation of this subunit. Mutations affecting G-proteins of the Gα_q/11_ class are almost systematically found in uveal melanoma. Approximately 90% of metastatic uveal melanomas are mutated for Gα_q_ or Gα_11_, affecting mainly glutamine 209 in both proteins, but also, to a lesser extent, arginine 183 [158,159]. *GNAQ*, which encodes Gα_q_, is mutated in 50 to 85% of non-epithelial melanocytic lesions, including blue nevi and leptomeningeal melanocytic neoplasms. Mutations in *GNA11*, which encodes Gα_11_, are more frequent in uveal melanomas [160]. These mutations are found at lower frequencies (~1–6%) in other types of cutaneous melanoma [110,161]. Q209L/P and R183C/Q mutations in GNAQ or GNA11 affect the GTPase domain [162] but only Q209L/P mutations have actually been characterized. These mutations reduce the GTPase activity of the Gα_q_ subunit and cause hyperactive signaling [162]. Regardless of the tumor context, the mutations are mutually exclusive [158] and are also mutually exclusive with BRAF and NRAS mutations. Of note, in uveal melanoma, mutations of CYSLTR2 (L129G), encoding a Gα_q/11_-coupled GPCR, are found in a mutually exclusive manner with Gα_q_ and Gα_11_ mutations in approximately 3% of patients. This mutation is also present in blue nevi [158,163]. This CYSLTR2^L129G^ mutation constitutively activates the receptor and thus downstream signaling. Depending on the murine models used and the cells targeted (neural crest cells or melanoblasts), embryonic Gα_q_^Q209L^ expression is able to induce a range of lesions from dermal hyperpigmentation to leptomeningeal melanocytoma, nevi, and dermal melanoma to malignant uveal melanomas with lung invasion [164,165]. Similarly, postnatal expression of Gα_11_^Q209L^ in melanocytes induces hyperpigmented melanocytic lesions in the uveal tract, skin, and leptomeninges that progress to melanoma with lung invasion [166]. Mice transplanted under the skin with Gαq mutated melanoma cells show inhibition of MAPK signaling and tumor growth following treatment with FR900359 [167]. To date, the inhibition of uveal melanoma with FR900359 appears to be more potent than inhibition with YM254890 [168]. The activation of Gα_q/11_ is a mechanism of resistance to MAPK pathway inhibition through the overexpression of c-Jun [169]. These data are consistent with the gain in resistance to MAPK inhibitors shown by activation of EDNRB [170]. Conversely, inhibition of Gα_q_ by YM-254890 resulted in inhibition of MAPK signaling, with evidence of a rebound after 24 h in xenograft experiments of uveal melanoma. Combined treatment with YM-254890 and a MEK inhibitor led to sustained MAPK inhibition and tumor shrinkage [171]. A combination of PKC and MEK1 inhibitors is currently under clinical evaluation for solid tumors harboring GNAQ/11 mutations or PRKS fusions (Phase I/II trials: NCT03947385).

The primary target of Gα_q/11_ subunits is phospholipase C beta (PLCβ) [172,173,174]. This subfamily is encoded by four genes (*PLCB1-4*) encoding seven proteins, all of which have two isoforms, except PLCβ3 [175]. All PLCβs are likely expressed in cutaneous melanoma [110], whereas, only the *PLCB2-4* genes are expressed in uveal melanoma [158]. PLCβs function by hydrolyzing membrane phosphatidylinositol-4,5-biphosphates (PIP2) into IP3 and DAG [176]. PLCB4 mutations occur in approximately 5% of uveal melanoma and are mutually exclusive with *GNAQ*, *GNA11*, and *CYSLTR2* mutations [177]. PLCβ4^D630Y^ mutations affect the Y domain of the catalytic core of PLC4 [177]. Their effect has not been precisely studied, but their exclusivity with the *GNAQ*, *GNA11*, and *CYSLTR2* mutations and PLCβ4 being downstream of these proteins suggest that PLCβ4 mutations have a similar effect on the oncogenicity of uveal melanomas [177]. Outside the context of melanoma, the other PLCs have often been shown to induce migration and/or to reduce the immune response [178,179]. 

One of the two secondary messengers produced by PLCs is DAG, which remains anchored in the membrane. Phorbol esters, such as 12-O-tetradeconoyl phorbol-13-acetate (TPA) and phorbol 12-myristate 13-acetate (PMA), are synthetic analogues of DAG. TPA is essential for melanocyte growth in vitro [180,181,182]. Interestingly, the effect of TPA on the proliferation of melanoma cell lines appears to be cell dependent [118,181,183,184,185]. PMA increases cell survival, invasion, and resistance to anoikis [184,186,187]. TPA and PMA activate the PKC and MAPK pathways [184]. The other secondary messenger produced by PLCs is IP3. The binding of IP3 to the IP3 receptor (IP3R) increases intracellular Ca^2+^ levels [188]. Calcium release induced by IP3 supports melanoma cell migration and invasion [189,190,191].

DAG and calcium activate protein kinase C by binding to the C1 and C2 domains of PKC, respectively [192]. There are nine genes that encode PKC: PKCα, PKCβ, PKCγ, PKCδ, PKCθ, PKCε, PKCη, PKCι, and PKCζ [192]. They are divided into three classes according to their activation mechanism. The classical PKCs (cPKCs) consist of PKCα, PKCβ, and PKCγ and are activated by calcium and DAGs. The novel PKCs (nPKCs) (PKCδ, PKCθ, PKCε, PKCη) are activated by DAGs alone. Atypical PKCs (aPKCs) (PKCι and PKCζ) are not activated by calcium or DAGs [192]. At least one PKC from each class is expressed in melanoma cells, except PKCγ [110,192]. PKC activity is regulated by the presence of their substrates and cofactors and their recruitment by scaffolding proteins, such as receptor for activated kinases C (RACK) or protein kinase A scaffolding protein 5 (AKAP5) [193,194,195,196]. PKC regulates invasion of melanoma cells but the various members have different effects: PKCα and PKCδ induce melanoma migration, invasion, and lung colonization [197]. Conversely, PKCβ decreases invasion and promotes cell differentiation and pigmentation [198,199]. *PRKCB* is frequently mutated and its expression is often lost in melanoma, but the impact of these mutations on PKCβ activity has not been evaluated [110,196]. Similarly, the effect of PKC on cell growth is dependent on the isotype [200,201,202].

Among the targets of PKC is Ras guanine-releasing protein 3 (RASGRP3), a guanine nucleotide exchange factor for RAS family proteins [166,200]. PKC phosphorylates RASGRP3 on Thr 133, which contributes to its activation in conjunction with DAG binding [203,204,205]. The inhibition of RASGRP3 induces the loss of GTP binding to RAS and thus its activity [166,200]. This decrease in activity is accompanied by a decrease in MAPK pathway activity and is associated with a decrease in cell proliferation [166]. This molecular mechanism may explain the activation of the MAPK pathway seen after PKC activation [86,206,207,208]. PKCε and PKCη are able to shunt the pharmacological inhibition of BRAF^V600E^, rendering melanoma cells resistant to these drugs [209]. Consistent with this finding, the use of PKC inhibitors inhibits the survival and migration of melanoma cells resistant to vemurafenib [210].

The activity of Yes-associated protein 1 (YAP1) is positively regulated by many GPCRs but negatively regulated by the Hippo pathway [211,212]. In uveal melanoma, YAP1 is activated by Gα_q_, inducing cell growth and survival [164,213,214]. Inhibition of YAP activity by verteporfin reduces tumor growth [213,214]. Gα_q_ activates the guanine nucleotide exchange factor TRIO, which in turn activates the small GTPases RhoA and Rac1 [213,215]. This likely has a dual action: (i) the activation of FAK, which in turn inhibits the LATS1/2 kinase of the Hippo pathway, inactivating the action of YAP1, and (ii) the direct activation of YAP1 by releasing the angiomotin transcription factor (AMOT) [213,216]. In uveal melanoma, the activation of Rho and Rac is linked to the activation of the MAP kinases JNK & p38 [215]. JNK and p38 phosphorylate c-Jun and induce the expression of AP-1 targets [217]. Phosphorylation of c-Jun induces cell proliferation via AP1 targets that regulate the cell cycle, such as cyclin D1, p53, p21cip1/Waf1, p19ARF, and p16 [217,218,219,220]. Activation of p38 and JNK are likely to be involved in mechanisms of resistance to MAPK inhibitors [218,219]. In cutaneous melanoma, the activation of YAP1 is required for cell invasion [221,222,223,224], as well as viability and resistance to anoikis [221,222,224]. Although the effect of YAP1 on invasion is clearly documented, its effect on melanoma growth is still debated [221,223]. YAP1 activity has also been shown to be associated with cell migration via regulation of the arp2/3 complex 3 [221,225]. The invasive phenotype of melanoma cells has been correlated with the activation signature of YAP [223]. In vivo, YAP1 activation induces the formation of very large numbers of metastases in the lungs after the injection of cells carrying the activating mutation of YAP1-5SA under the skin of mice [223]. Lung colonization after the injection of cells into the tails of mice also decreases when YAP1 levels are genetically decreased [222,224]. Most interestingly, inhibition of the Hippo pathway replicates this effect, favoring lung colonization [226]. The pro-invasive action of YAP1 is mediated through the transcription of a number of its targets, such as CCN1, AXL, and THBS1 [223]. These targets are well known in melanoma. AXL expression is associated with tumor growth and cell invasion and migration and has also been shown to be associated with resistance to MAPK inhibitors [227,228,229]. CCN1 has been shown to be associated with increased metastatic potential and angiogenesis [150,230]. Genetic inhibition of THBS1 is associated with decreased cell invasion [231,232]. YAP1 requires the transcriptional cofactors TEAD1-4 to bind to its targets. Genetic inhibition of TEAD1-4 recapitulates the in vitro effects of YAP1 on invasion, with a clear decrease in the invasive capacity of melanoma cells [233]. TEAD1-4 are also involved in resistance to MAPK inhibitors and inhibition of all four TEADs sensitizes cells to these inhibitors [233].

### 3.3. Signaling via Gα_12/13_

The induction of PAR1 by its ligand activates Gα_12/13_. It has to be noted that PAR1 induces both the IP3/DAG and Gα_12/13_ pathways. The third major pathway of GPCR signaling involved in melanoma is the pathway involving Gα_12/13_ [100]. This class of G subunits has two members, Gα_12_ and Gα_13_, encoded by the ***GNA12*** and ***GNA13*** genes, respectively. The expression of these genes is ubiquitous [100,110,115]. Activation of these subunits activates Rho-GEFs, such as leukemia-associated Rho-GEF (LARG) or p115 Rho-GEF [234,235]. Activation is achieved by the attachment of the G12/13 subunit to the RH domain of Rho-GEFs [236]. Once activated, Rho-GEFs induce RhoA activation [237,238,239,240]. Rho is a converging point for Gα_12/13_ and Gα_q/11_ signaling [239]. Signaling induced downstream occurs through the activation of YAP1, as described above [212]. Gα_12/13_ signaling has been poorly analyzed in the context of melanoma. PAR1 and 2 receptors, coupled to Gα_12/13_ proteins, are expressed in melanoma [241]. Activation of PAR1 receptors by its ligand TRAP6 induces the activation of YAP1 in a HEK293A cell model [242]. YAP1 activation subsequently leads to activation of RhoA and inhibition of LATS1 [242]. Similar to Gα_q/11_ signaling, Gα_12/13_ signaling appears to increase invasion and migration while not altering cell growth [243,244]. However, these results were obtained in breast and prostate cancer and need to be confirmed in melanoma. Of note, activation of LPA receptors (LPA1-LPA6), coupled to G_12/13_ receptors, is still poorly documented in melanoma, although it has been reported to enhance chemoresistance and increase the survival of melanoma cells in vitro [245]. 

### 3.4. Signaling via WNT/β-Catenin

The induction of FZD7 by its ligand activates the **WNT/β-catenin** signaling pathway. WNT ligands activate three intracellular pathways: the canonical WNT/β-catenin, WNT/Ca^++^ and WNT/PCP. Only the canonical WNT/β-catenin pathway will be described in this chapter. WNT/Ca^++^ corresponds to the Gα_q/11_ pathway already described in 3.2. and the WNT/PCP pathway has not been studied in melanoma. 

The WNT/β-catenin pathway has been widely studied and reviewed. Here, we provide the current knowledge of this pathway in melanoma, especially through the activity of β-catenin, which is encoded by *CTNNB1* [68,246]. 

Mutations in the *CTNNB1* gene are rare: around 3% in melanoma except for the deep penetrating nevus (DPN) that harbors 90% of activating mutations in *CTNNB1*. As such, activated β-catenin is a marker caracterizing this specific melanocytoma/melanoma type from other cutaneous melanocytic tumors [247,248]. However, cytoplasmic or nuclear localization of β-catenin, indicating activation of the pathway by other mechanisms other than mutation in its gene, was found in 20–30% of human melanoma [249,250]. The canonical WNT pathway is activated only in response to the formation of a complex containing WNT, FZD, and LRP. WNT proteins are difficult to purify in an active form and only a few antibodies are available for their detection. The WNT proteins most studied in the context of β-catenin activation in melanocytes/melanoma are WNT1 and WNT3a. WNT5a has different roles and acts as an antagonist or agonist of the canonical WNT/β-catenin pathway, depending on the cellular context. WNT proteins are subject to post-translational modifications, including glycosylation and lipid modifications. Acylation on conserved serine and cysteine residues is required for WNT secretion and efficient binding to the Frizzled receptor [251,252]. In the basal state, Axin protein interacts through distinct domains with GSK-3, CK1α, APC, and β-catenin and is considered to be the limiting component of the β-catenin destruction complex [253,254]. Modulation of its levels would therefore be an effective way to regulate β-catenin destruction. APC is a large protein that interacts with both β-catenin and Axin. It contains three Axin-binding domains, interspaced between armadillo repeat domains (ARMs), which bind to β-catenin. β-catenin is sequentially phosphorylated by CK1α and GSK-3 on serines (S) and a threonine (T) (S45, T41, S37, and S33) in the N-terminal region of the protein, resulting in its interaction with and ubiquitination by β-TRCP1 before being degraded by the proteasome. 

Binding of the WNT ligand leads to the dimerization of Frizzled with the coreceptor LRP5/6. This dimerization results in a conformational change of the receptors, leading to relocalization of the degradation complex to the cell membrane under the double interaction of Axin with DVL (itself associated with Frizzled) and with the cytoplasmic end of LRP. Such membrane relocalization decreases the activity of the destruction complex, such that the amount of unphosphorylated cytoplasmic β-catenin rapidly increases. The stabilization of cytoplasmic β-catenin results in an increase in nuclear β-catenin. The balance between the amount of cytoplasmic and nuclear β-catenin is dynamic, resulting from multiple mechanisms of transport and retention between the two compartments. In the nucleus, β-catenin binds to the T-cell factor (TCF)/lymphoid enhancer-binding factor (LEF) family of transcription factors, which themselves are already associated with DNA. In the absence of β-catenin, TCF factors interact with transcriptional co-repressors of the Groucho/transducin-like enhancer of split (TLE) family and repress the expression of their target genes. Nuclear accumulation of β-catenin leads to the association of TCF with β-catenin, resulting in dissociation from Groucho/TLE1 and allowing the recruitment of other coactivators for transcriptional activation through its C-terminal transcriptional activation domain. Many transcription factors outside the TCF/LEF family of transcription factors have been reported to be capable of associating with β-catenin to activate or repress transcription [255]. In addition, it has to be noted that alternative pathways activate β-catenin independently of WNTs [256].

In melanocytes/melanoma, MITF interacts with β-catenin and redirects β-catenin-mediated transcriptional activity from canonical Wnt/β-target genes to specific MITF target genes to activate their transcription [257]. For instance, in mutated β-catenin melanoblasts, stabilised form of β-catenin increases Mitf-M levels, which may interfere with β-catenin transactivation, inhibiting the activation of Myc and CyclinD1, therefore reducing proliferation [74]. During melanocyte establishment, β-catenin and MITF-M levels are likely maintained within a very narrow range, with any reduction or increase, such as those observed in the bcat mutants, altering melanoblast proliferation. Other levels of cross-signaling between the WNT/β-catenin pathway and MITF have been described. For example, it has been shown that β-catenin/TCF transcriptionally upregulates MITF-M expression [72]. It has also been shown that MITF-M binds to and upregulates its own promoter through a direct interaction with LEF1 [258]. It would appear to be difficult to target β-catenin in melanoma without affecting MITF expression and/or activity. MITF could potentially be an attractive target for melanoma therapy but the drug-targeting of MITF is highly challenging. As mentioned already, MITF is considered to be the “master gene” of melanocyte differentiation and has an essential role in the proliferation, survival, senescence, migration, invasion, DNA repair, and metabolism of melanoma cells [52,259]. Mitf expression is regulated by multiple signaling pathways outside of the canonical Wnt/β-catenin pathway, such as the cAMP/CREB, YAP1/PAX3, TGFβ/GLI2, and TNF/NFκB pathways, and by transcription factors, such as SOX10 and BRN2, themselves regulated by multiple pathways in melanoma. The basic concept in melanoma is that the proliferative and invasive states are defined, in part, by the high level/activity of MITF and low level/activity MITF, respectively. High and low MITF level/activity co-exist in melanoma tumors and the switch in MITF expression (high and low) is reversible and responsible for melanoma heterogeneity and plasticity. MITF is also involved in the resistance to BRAF inhibitors. One current view for the therapeutic strategy is to increase MITF levels and therefore those of melanoma antigens, such as MART-1 and GP-100, to increase the recognition of melanoma cells by T cells and improve the immune response [260]. In any case, therapeutic strategies have to address the versatility and heterogeneity of melanoma cells.

### 3.5. Signaling via Gβ/Gγ Subunits

All GPCRs activate the Gβ/Gγ signaling pathways, but this pathway remains poorly studied. The activation of GPCRs mainly induces the activation of Gα subunits, but also that of the Gβ/Gγ complex. There are five Gβ subunits (Gβ_1-5_) in humans encoded by the *GNB1-5* genes. All except *GNB5* are expressed in melanoma [100]. There are 16 Gγ subunits encoded by the *GNG1-16* genes, of which only Gγ_2_, Gγ_4_, Gγ_5_, Gγ_6_, Gγ_7_, Gγ_10_, Gγ_11_, and Gγ_12_ are expressed in melanoma [100,110,115]. The significance of Gβ/Gγ signaling in melanoma has not yet been fully assessed. Gβ/Gγ subunits activate the PI3K signaling pathway. This activation is either direct or indirect via calcium release [261,262,263,264]. Classically, PI3K activation induces AKT activation and increases cell survival [265]. Also, Gβ/Gγ were shown to inhibit melanoma migration in vitro through EPAC inhibition [261]. Given the known importance of the PI3K/AKT pathways, as well as the role of migration in melanoma, its regulation by Gβ/Gγ subunits needs to be further analyzed.

### 3.6. β-Arrestin Signaling

All GPCRs activate **β-arrestin** biased signaling, but this remains poorly studied in melanoma. At the end of the 1990s, it was observed that Src family tyrosine kinases are recruited by β-arrestins (encoded by *ARRB1-2* genes) to the adrenergic receptor 2a, a member of the GPCR family [266]. Surprisingly, the binding of protein kinases induced activation of the MAPK/ERK pathway only if the receptor was internalized [266,267]. Other observations in the mid-2000s showed that such signaling was non-canonical and independent of G-proteins. Indeed, the activation of GPCRs activates three types of proteins: G-proteins, GPCR protein kinases (GRKs), and arrestins. GRKs and arrestins are the most important elements involved in the termination of GPCR activation. GRKs phosphorylate the receptor on its C-terminal residues, which prevents the activation of G-proteins [268]. Such phosphorylation recruits the non-visual arrestins, β-arrestin 1 & 2, which in turn recruit clathrin, resulting in receptor internalization by clathrin coated-pits [268,269,270]. Depending on the affinity of the GPCR/β-arrestin complex, receptors can be recycled or degraded in the proteasome [271]. In early endosomes, the GPCR/β-arrestin complex is able to form a signalosome by recruiting signaling proteins, such as members of the MAPK pathway [270,272,273]. As a result, several pathways can be activated, such as the MAPK/ERK and Src pathways [266,274], AKT [275], MAPK/JNK [276], MAPK/p38 [277], or PDEs [278].

GRK-arrestin signaling has been poorly studied in melanoma. β-arrestin2 is able to bias MC1R signaling by promoting activation of the MAPK pathway towards that of cAMP-dependent signaling [279]. A transcript of the MC1R (MC1R-203) naturally promotes such biased signaling toward the MAPK pathway [280]. Mutants of metabolic glutamate receptors (mGluR3^G848E^) can promote such biased signaling, characterized by prolonged internalization of the receptor [281]. This mutation is found in rare cases of cutaneous melanoma (<1%) [110]. Conversely, in uveal melanoma, the CYSLTR2^L129Q^ mutation forces signaling via Gα_q/11_ and disfavors β-arrestin-biased signaling [282]. GPCR signaling is certainly much more complex than what is presented here: their interactions with GRKs and arrestins and the dynamics of their desensitization add another level of complexity that needs to be investigated in the future.

## 4. Perspectives of Targeting GPCRs in Melanoma

The molecular and cellular consequences of the activation of GPCRs in melanoma clearly indicate that GPCRs could be interesting targets for therapy. Targeting GPCRs has the advantage of seeking readily available membrane molecules instead of signaling proteins with molecules that need to cross the plasma membrane without deteriorating the intracellular environment and/or endocytotic vesicles and lysosomes [283,284,285].

### 4.1. Limitation of Available Tools

Large-scale sequencing conducted by the cancer genome atlas consortium (TCGA) has shown that approximately 92% of the melanoma patients tested had at least one nonsense or missense mutation of at least one non-olfactory GPCR [110,286]. Each melanoma patient has an average of 10 mutated GPCRs [110,286]. The functions of the vast majority of the receptors and associated mutations remain uncharacterized. The most frequent GPCR recurrent mutation in melanoma is GPR139^R217C^, but this mutation does not exceed 1% of the patients and may have a driver function. A large number of mutations have been found in various GPCRs, but can be considered as passengers since none of these mutations are recurrent in melanoma. Indeed, the Adhesion G Protein-Coupled Receptor V1 (*ADGRV1*) is mutated in approximately 30% of patients, but only a few recurrent mutations are found. The high number of mutations in ADGRV1 is most likely associated with the size of its cDNA (19,557 nucleotides) [110,286]. These data suggest that if we cannot offer patients a personalized and targeted therapy (one patient = one mutation = one drug), patients could be grouped by activated signaling for therapy that is still targeted but less personalized. This less direct choice of therapy would consider the positioning of the GPCR within the signaling pathways and target the downstream node(s) [287]. Such a therapeutic option would also be attractive in the setting of patients who do not have mutated GPCRs but rather mutations in downstream signaling elements. 

Receptor activation may result by a mechanism independent of the presence of a mutations in the gene. This could be related to overexpression or *de novo* expression of a receptor, which would induce a higher basal level of receptor activity, or activation of the receptor by the production of its ligand in the environment [288,289,290]. Binding of the ligand to the receptor may occur in the primary melanoma in an autocrine/paracrine mode, with ligand production by the melanoma cell or by cells in the microenvironment [291]. For example, activation of EDNRB by ET-1 secreted by surrounding melanoma cells induces reactivation of the MAPK pathway after BRAFi treatment [292]. In this model, ET is secreted by melanoma cells and activates EDNRB in an autocrine and/or paracrine manner [292]. Alternatively, ligand production may only occur at the metastatic site and thus only affect metastasis formation and not melanoma initiation [293,294]. Ligand production may attract tumor cells into the target tissue by chemoattraction, promote cell survival and/or proliferation, or induce resistance to drugs. In this perspective, ET-1 is highly expressed in the lungs and can promote the colonization of melanoma cells that express ENDRB in this organ. The reactivation of MAPK pathways by expression of EDNRB may occur at primary sites, as well as in distant organs. Ligand-dependent tissue expression can be observed, in particular, in the lungs, where only 50 highly expressed ligands are found [114,295,296,297,298]. An important limitation is our lack of knowledge about the level of gene expression in melanoma cells that colonize distant organs. Transcriptomic analyses would be extremely useful in determining which GPCRs are expressed in melanoma in distant metastases. The correlation of GPCR expression data in melanoma at sites of metastasis with expression in the target tissue of ligands for these GPCRs would allow the selection of potential receptors of interest.

An alternative to transcriptional analysis would be the identification of GPCRs at metastatic sites on the basis of protein expression. However, the identification of GPCRs by immunolabeling is difficult, due to the small exposed area of extracellular epitopes and very high conformational variability. Conventional mass spectrometry is also challenging, as receptors are not readily isolatable from membranes [299,300,301].

The improvement of antibody-isolation technologies associated with our knowledge of GPCRs will lead to the generation of new, more selective antibodies. The development of nanobodies appears to be promising for the detection and targeting of GPCRs [302]. The major limitation in the high-throughput identification of GPCRs via mass spectrometry is the depletion of GPCRs from mass spectrometry samples. This bias can be avoided by performing surfaceomes, which will significantly enrich samples for membrane glycoproteins and thus potentially reveal the expression of GPCRs [301,303,304]. Finally, GPCRs can also be indirectly identified by analyzing cell-binding ligands rather than the receptors directly [305].

Data mining of the literature generates databases, such as TCGA. To date, current databases have been extremely useful for the identification of driver mutations and prognostic biomarkers in melanoma. However, they are composed of 80% primary tumors or skin or lymphatic metastases [110,286], whereas visceral, bone, and nervous system melanoma metastases are poorly represented. The presence of characteristic and highly aggressive mutations in distant metastasis may be hidden by the small sample size. Furthermore, current databases were generated before the generalization of current treatments and the samples constituting the large databases came from patients who were naive to any treatment with MAPK or checkpoint inhibitors. The changes in genetic/epigenetic expression due to such treatments could involve the activation of GPCR expression or the selection of subclones in which GPCR signaling is activated. The generation of databases enriched with samples from distant metastases and patients treated with MAPK and immune checkpoint inhibitors would be of great interest to the scientific community.

Large-scale high-throughput screening using RNA interference or CRISPR-Cas9 could be performed to identify the GPCR-dependence of melanoma cells for growth or invasion. Such screens can also be performed in vivo. To evaluate the role of specific GPCRs, transplantation of genetically modified cells into animals would reveal their roles in internal organ colonization and in melanoma progression [306]. Such screening could also be performed by generating transgenic animal models, but the experiments would become extremely complex, time consuming and expensive. Zebrafish may be a suitable model for these types of studies. Indeed, it is relatively easy to generate and maintain large groups of transgenic animals [307,308,309,310]. Furthermore, zebrafish express many of the GPCRs expressed in humans, as well as their signaling machinery [311,312]. In addition, there are already many melanoma models that could be used to test genetic modifications in melanoma-producing animals, for example, to assess the pro-metastatic effect [313,314]. Particular attention should be paid to studies based on in vivo cell-injections. Indeed, the type of injection will determine the preferential location of metastasis formation. For example, in mice, tail-vein injections are more likely to result in lung metastasis formation, whereas intrasplenic injections are more likely to result in liver metastases [315]. Thus, the use of a particular type of injection may mask the action of a GPCR on metastasis formation in a tissue not targeted by the injection method. The study of the tissue expression profile of GPCR ligands and the use of an appropriate method can address this concern.

### 4.2. Novel Structures and Drug Design Approaches

To efficiently target GPCRs, two parallel and complementary approaches are used: binding of compounds (including small molecules, antibodies and radiotherapies) to GRPRs, and structure determination of GRPRs. The structure of GPCR can be elucidated using X-ray crystallography, Cryo-EM, NMR and artificial intelligence. 

A better knowledge of GPCR structure and molecular ligand-receptor interactions is critical for structure-based molecule design and the design of new receptor-activating agonist or antagonist molecules [316]. Such structural knowledge needs to be as accurate as possible and is currently acquired by two main experimental methods and one predictive method [316,317].

The first three-dimensional structure of a GPCR, rhodopsin, was solved by X-ray crystallization in 2000 [318]. This method requires obtaining stable crystals of the receptor extracted from its membrane in solution, which has been a major obstacle to structure determination [319]. This problem has been solved by modifying GPCRs to make them more stable while retaining their activity [319,320,321,322]. This can be achieved, for example, by fusing the T4 lysozyme protein as a replacement for intracellular loop 3 [321] or by mutating the receptor to thermostabilize it [320]. A new technique for crystal structure resolution has recently been developed: X-ray-free electron laser (XFEL) crystallography. The use of lasers reduces the size of the crystals required and, therefore, increases their stability [323,324].

Crystallization has been completed by cryo-electronmicroscopy (Cryo-EM). Significant improvements in detectors and structure determination algorithms have enabled the resolution of structures at the particle scale [325,326,327]. Cryo-EM does not require the formation of crystals, as the receptors are directly vitrified after purification, but requires more computational time to solve the structure, resulting in lower resolution [317,328,329,330]. These various constraints and the relative novelty of these techniques mean that, currently, the structure of only 20% of non-olfactory GPCRs has been solved [328]. Nuclear magnetic resonance (NMR) allows the acquisition of dynamic data and thus complements the data from the fixed structures of crystallography and Cryo-EM [331]. 

More recently, advances in artificial intelligence have made it possible to approach structures using machine learning. Protein structure prediction was revolutionized in 2021 by the publication of the algorithms *AlphaFold2* and *RoseTTAfold*, which generated highly accurate three-dimensional structures of any protein of interest [332,333,334]. These algorithms are completed by GPCR-specific algorithms, such as the already developed arsenal, which is specific to the GPCR field [335,336,337,338]. Nonetheless, although these algorithms all regularly perform well, they can still be improved. GPCRs differ, particularly in their loops, which results in a deficit for machine-learning algorithms and thus low confidence in the models, often producing false predictions [335,339]. Generating and publishing more receptor structures will increase both direct knowledge about receptors and knowledge about other receptors via increased substrates for algorithms and in silico prediction. Predictions of molecular anchors in receptors can also be made by site-directed mutagenesis (SDM) studies to find amino acids that interact with known ligands of the receptor [340,341]. Crystallization and SDM techniques can be used synergistically to better understand the binding of the molecule to the receptor and produce much more refined and/or efficient molecules [340,341,342].

For the development of an effective therapy, stable, receptor-specific agonists or antagonists must be found. The screening of drug libraries is performed to find molecular scaffolds capable of binding to receptors and modulating their activity [343]. Screening can be performed using affinity assays by assessing the ability of molecules to bind to the target as purified proteins, on whole cells expressing the target, or on isolated membranes [344,345,346]. Purified proteins are free of binding to secondary targets, but the ability of the molecules to cross a membrane is not analyzed. Moreover, the presence of the membrane is often necessary for receptor stability [345].

Affinity tests are based on the ability of molecules to displace—and replace on the receptor—a reference ligand known to bind the receptor. Detection is either by radio-labeling of the reference ligand, regularly labeled with iodine-125 or tritium [347,348,349] or by fluorescence resonance energy transfer (FRET), in which both the reference ligand and the receptor are bound to a fluorophore that allows resonance from one molecule to the other [350,351]. Such screening is only able to identify ligands that bind to the same site as the competitor. To overcome this bias, the analysis of the activity of various elements of GPCR signaling, such as IP1 concentration for the Gα_q_ pathway or cAMP for the Gα_s_ pathway, can be jointly performed on a large scale [352,353]. The activity of molecules on downstream G-protein signaling pathways can also be assessed using kinase assays from protein extracts or purified protein or by FRET [354,355,356]. These screens can be virtually performed with high efficiency if the receptor structure is known [357,358,359]. The hits that are found are generally of low affinity for their targets and require optimization to be usable [343]. They will then need to undergo pharmaco-modulation that will be directed through structural data and docking algorithms to result in lead generation [360,361,362]. These leads will have to be tested in vivo to verify their pharmacokinetic parameters (absorption, distribution, metabolism, elimination) at the risk of obtaining only molecules incapable of producing an effect in vivo and condemned to be used only as an in vitro tool [363,364]. Molecules identified as leads can be used to verify the efficacy of these molecules on the cellular and molecular processes of melanoma progression, as well as on the formation of metastases and resistance to treatment.

In addition to small chemical molecules, GPCRs can be targeted by monoclonal antibodies. Several have been generated and approved as therapeutic targets, such as mogamulizumab, a humanized antibody against chemokine receptor type 4. The advantage of using monoclonal antibodies in treating diseases is notable because they have a long half-life.

Finally, targeted radiotherapy is currently being developed and constitutes a promising strategy to target GPCRs in cancer. The radioactivity is delivered to a specific tumor by means of a systemic injection. For melanoma treatment, targeted treatment can be achieved by labeling small molecules, such as melanin ligands, peptides that recognize a specific receptor (MC1R), or antibodies (anti-melanin, anti-GD3) [365,366]. The efficacy of targeted radiotherapy depends on the dose delivered, which in turn depends on the radionuclide used and the time that the labeled compound remains associated with the target. Targeted radiotherapy is highly relevant for the treatment of disseminated lesions and overcoming tumor heterogeneity through cross-fire irradiation with β-radionuclides characterized by a decay spectrum of between a few nanometers and 2 mm. The reception of adequate doses of radiation from neighboring receptor-expressing cells can kill tumor cells lacking the targeted receptor in the tumor. This property is particularly important for melanoma, in which gene expression is often heterogeneous. Targeted radiotherapy of the somatostatin receptor is used for the clinical treatment of neuroendocrine tumors. The same strategy could be adapted for targeting GPCRs in melanoma. 

## 5. Conclusions

Of the eight GPCRs shown to be involved in pigmentation, only three have been studied in the context of melanoma (EDNRB, MC1R, and GRM1). The remaining five receptors (DRD2, FZD4, GPR143, GPR161, and SMO) may be of interest in the context of melanoma, but they have not been studied yet. Conversely, it would be relevant to evaluate the importance of the eight receptors described only in a melanoma context (FZD7, GPER1, GRM3, GRM5, PAR1/F2R, CXCR4, CCR7, and CCR10), for their implication in the melanocyte lineage as a whole, whether during embryonic development, melanocyte homeostasis/renewal and/or melanogenesis. The identification of completely novel GPCRs, important in melanomagenesis, will require detailed studies of primary and/or metastatic tumors through the development of new and more powerful analysis tools as their detection may escape current RNAseq and proteomic techniques.

The low number of GPCR mutations and their modification of expression/activity often dependent on temporal, tissue and molecular contexts render their studies in vivo essential to precisely define their roles. The creation of novel Omics databases generated from distant melanoma metastases treated with immune checkpoint or/and MAPK inhibitors would reveal novel GPCR players involved in resistance and metastasis. Once such GPCR target(s) would be identified and characterized, small molecules will have to be developed to directly target it/them. These molecules can be screened from chemical libraries or designed according to their structure, based on their crystal or artificial intelligence using Alphafold for instance. 

Prior to 2010, surgery remained the primary treatment option for resectable melanoma, with eventual chemotherapies, although these treatments did not provide a survival benefit for patients when metastasis arose. Over the past decade, effective therapeutic approaches such as immunotherapies and targeted therapies, alone or in combination, have significantly changed the prognosis of patients with advanced melanoma. However, despite the success of these treatments, their long-term efficacy remains limited by mechanisms of acquired resistance. In this review, we have highlighted the involvement of GRPRs in the initiation and progression of melanoma and mentioned their therapeutic evaluation in preclinical or clinical trials. Most of these studies are in the early stages of clinical development, being studied in phase I or phase I/II. The success of these trials will depend on understanding the activity of GPCRs and the quality/stability of the drug used in clinical development. GPCRs signal through one or more G-proteins and thus can regulate multiple signaling pathways simultaneously. Therefore, targeting GPCR activities and downstream signaling nodes may simultaneously inhibit several key signaling pathways in melanomagenesis. This strategy should be more effective (multi-pathway inhibition) in a greater proportion of patients (BRAF or NRAS mutated) than existing targeted therapies, and should minimize the development of cancer cell resistance. A better understanding of the specific function of GPCRs in melanoma will certainly lead to innovative therapy in the future and will open new avenues for the treatment of melanoma.

## Figures and Tables

**Figure 1 cancers-14-00706-f001:**
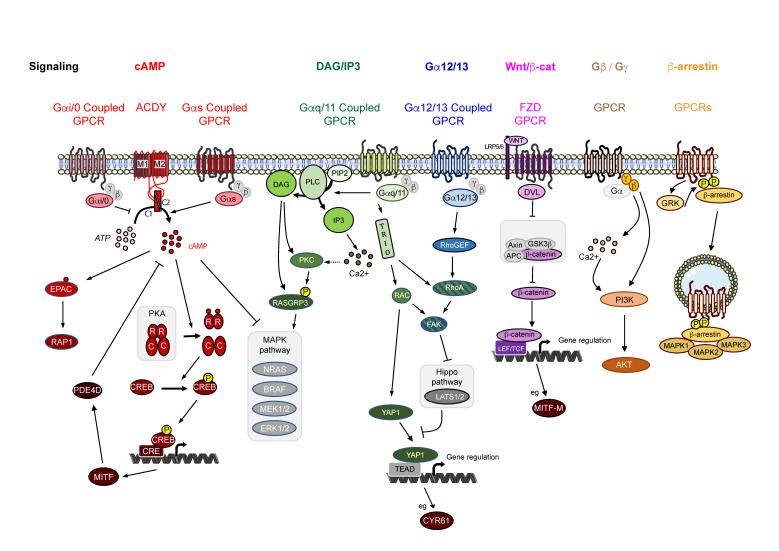
GPCR signaling pathways. Note that the Gα_q/11_ et Gα_12/13_ G-proteins share a common downstream signaling intermediate—YAP1.

**Table 1 cancers-14-00706-t001:** GPCR and pigmentation GPCRs involved in pigmentary phenotypes in mouse and human.

*Gene* *Cyto.loc*	Receptor Name	Signaling	Mouse Model	Human Disease	Function(s)	References
*DRD2* *11q23.2*	Dopamine D2	Gαs	Hyperpigmentation eg. Drd2^tm1Ebo^/Drd2^tm1Ebo^		αMSH synthesis	[7]
*EDNRB* *13q22.3*	Endothelin B	Gαq	Hypopigmentation eg. Ednrb^s-l^/Ednrb^s-l^	WS4A (#277580) ABCD syndrome (#600501)	Proliferation Survival DNA repair	[8]
*FZD4* *11q14.2*	Frizzled 4	β-catenin	Hypopigmentation eg. Fzd4^tm1Nat^/Fzd4^tm1Nat^			[9]
*GPR143* *Xp22.2*	Protein G143	Gαq	Eye hypopigmentation eg. Gpr143^tm1Inc^/Y	OA1 (#300808, #300500)	Melanogenesis	[10]
*GPR161* *1q24.2*	Protein G141	Gαs	White belly spot eg. Gpr161^vl^/Gpr161^vl^		Shh negative regulator	[11]
*GRM1* *6q24.3*	Glutamate receptor metabotropic 1	Gαq	Hyperpigmentation eg. Tg(Dct-Grm1)ESzc		Proliferation Survival	[12]
*MC1R* *16q24.3*	Melanocortin 1 receptor	Gαs	Yellow coat Mc1r^e^/Mc1r^e^	Light skin, red/blond hair	Switch eu/pheomelanin DNA repair	[13]
*SMO* *7q32.1*	Smoothened	Gli		Striped depigmentation Curry-Jones syndrome (#601707)		[14]

The major signaling is indicated for each protein as well as the most characterized mouse mutant. For some of the proteins a human pathology is associated, the OMIM link is indicated (#). The reference is indicated in brackets in the last column. Cyto.loc. = cytogenic location, Ocular albinism = OA, Waardenburg syndrome = WS, Albinism, black lock, cell migration disorder of the neurocytes of the gut = ABCD.

**Table 2 cancers-14-00706-t002:** GPCR and melanoma.

*Gene* *Cyto.loc*	Receptor Name	Signaling	Animal Model	Melanomagenesis	Function(s)	References
*EDNRB* *13q22.3*	Endothelin B	Gα_q_	Mitf-cre/+; Rosa-fs-GNAQ^Q209L/+^; Ednrb^F/F^	initiation progression	Proliferation Invasion DNA repair	[21] [22]
*MC1R* *16q24.3*	Melanocortin 1	Gα_s_	Tyr::CreER^T2^/°; Braf^V600E^; MC1R^e/e^	initiation	Proliferation DNA repair	[23] [24]
*FZD7* *2q33.1*	Frizzled 7	β-catenin	Xen., A375P, WM1361 shRNA-FZD7	progression	Proliferation	[25] [26]
*GPER1* *7p22.3*	G-protein-coupled estrogen receptor	Gα_s_	Xen., NHEM with BRAF^V600E;^ p53^R248W^; CDK4^R24C^; hTERT	initiation	Differentiation	[27]
*GRM1* *6q24.3*	Glutamate receptor metabotropic 1	Gα_q_	Dct::Grm1	initiation progression	Proliferation Invasion	[12] [28]
*GRM3* *7q21.11/12*	Glutamate receptor metabotropic 3	Gα_q_	Xen., A375 GRM3 S610L, E767K, or E870K;	initiation progression	Proliferation Migration	[29]
*GRM5* *11q14.2/3*	Glutamate receptor metabotropic 5	Gα_q_	Tyrp1::Grm5	initiation progression	Proliferation Invasion	[30]
*PAR1/F2R* *5q13.3*	Coagulation factor II Thrombin	Gα_q_	Xen. A375 shRNA-PAR1	progression:	Proliferation	[31]
*CXCR4* *2q22.1*	CXC motif chemokine 4	Gα_i/0_	Allograft B16 Ectopic expression Blocking peptide T22	progression	Attraction Growth	[32] [33]
*CCR7* *17q21.2*	CC motif chemokine 7	Gα_i/0_	Allograft B16 Ectopic expression Neutral. anti-CCL21 ab	progression	Attraction Growth	[34]
*CCR10* *17q21.2*	CC motif chemokine 10	Gα_i/0_	Allograft B16 Ectopic expression	progression	Immune invasion Apop. resistance	[35]

GPCRs involved in melanoma from transgenic mice model or cell grafting or injection. The main associated signaling pathway is indicated as well as the cellular processes induced by the receptor activation. The reference is indicated in brackets in the last column. Cyto.loc. = cytogenic location, ab = Antibody, Xen. = Xenograft, Neutral. = Neutralizing, Apop. = Apoptosis.

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
