# Peer review of "Targeting GPCRs and Their Signaling as a Therapeutic Option in Melanoma"

_cancers, 2022, doi:10.3390/cancers14030706_

Round 1

Reviewer 1 Report

Dear Editors, 

The review by Raymond J et al, is well documented and gives the reader a complete overview of the GPCR signaling in the melanocytic lineage. The manuscript is well written and to my knowledge it is the first review of that comprehensivley 

Specifics: 
Strength of the paper:
- first broad overview of GPCRs involvement in melanomagenesis
- first comprehensive review on GPCR mutated in melanoma

Minor comments
-line164: the following sentence" Recessive mutants at either of these loci " does not sound well. They author could consider using the word mutation instead of mutant. 
-Line 250: it is stated "MC1R is inactivated in people with red hair, due to mutation(s)". The author might consider speaking here about polymorphisms rather than mutations. 
-line 250: MC1R is not solely responsible for red hair phenotype. In the case of mutation of POMC, this phenotype is also observe.
In the Wnt section, author might consider adding a sentence or two on deep penetrating nevi and nuclear b-catenin. 

Author Response

Dear Editors, 

The review by Raymond J et al, is well documented and gives the reader a complete overview of the GPCR signaling in the melanocytic lineage. The manuscript is well written and to my knowledge it is the first review of that comprehensivley 

Specifics: 
Strength of the paper:
- first broad overview of GPCRs involvement in melanomagenesis
- first comprehensive review on GPCR mutated in melanoma

Minor comments
1. -line164: the following sentence" Recessive mutants at either of these loci " does not sound well. They author could consider using the word mutation instead of mutant. 

A1. “mutants” was replaced by “mutations” new line 167

  1. -Line 250: it is stated "MC1R is inactivated in people with red hair, due to mutation(s)". The author might consider speaking here about polymorphisms rather than mutations. 

A2. “mutations” was replaced by “polymorphisms” new line 255

  1. -line 250: MC1R is not solely responsible for red hair phenotype. In the case of mutation of POMC, this phenotype is also observe.

A3. We added a sentence line 259 “Similarly, patients with mutations in proopiomelanocortin (POMC) genes – encoding the precursor of aMSH – have red hairs [40]”

  1. In the Wnt section, author might consider adding a sentence or two on deep penetrating nevi and nuclear b-catenin. 

A4. We added a sentence. see lines 837-842.

Reviewer 2 Report

In this review article the authors have comprehensively discussed the role of GPCR signalling in melanomagenesis.

  1. Line 294-301: The authors must cite suitable references.
  2. Figure 1, LAT1/2: the authors must correct the typographical error. 
  3. It will be better, if the authors authors can add a brief paragraph on a clinical perspective.

Author Response

In this review article the authors have comprehensively discussed the role of GPCR signalling in melanomagenesis.

  1. Line 294-301: The authors must cite suitable references.

A1. We added 3 references. See lines 303 and 305.

  1. Figure 1, LAT1/2: the authors must correct the typographical error. 

A2. We corrected LAT1/2 -> LATS1/2 in figure 1

  1. It will be better, if the authors authors can add a brief paragraph on a clinical perspective.

A3.We added at the end of the review a paragraph on clinical perspective.

Reviewer 3 Report

The review titled "Targeting GPCRs & their signaling as a therapeutic option in  melanoma" is quite interesting. As widely known GPCRs mediates a wide range of biological functions and regulate intercellular signaling cascades.  The role of GPCRs as therapeutic option for the treatment of melanoma is nicely discussed in this review. The summarization of GPCR signaling pathways in Figure 1 is much appreciated. The description of each signaling pathway in detail will be of interest to a general reader as well. Overall, this an organized and well written review.

However, the Author Contributions, Acknowledgments and Conflicts of Interest sections should be revised. 

Author Response

The review titled "Targeting GPCRs & their signaling as a therapeutic option in  melanoma" is quite interesting. As widely known GPCRs mediates a wide range of biological functions and regulate intercellular signaling cascades.  The role of GPCRs as therapeutic option for the treatment of melanoma is nicely discussed in this review. The summarization of GPCR signaling pathways in Figure 1 is much appreciated. The description of each signaling pathway in detail will be of interest to a general reader as well. Overall, this an organized and well written review.

  1. However, the Author Contributions, Acknowledgments and Conflicts of Interest sections should be revised. 

A1. We revised the Author Contributions, Acknowledgments and Conflicts of Interest sections